



# Data-Driven Surrogate Models for Real-Time Fatigue Monitoring of Chain Mooring Lines in Floating Wind Turbines

Azélice Ludot[1,2], Thor Heine Snedker[1], Athanasios Kolios[2], and Ilmas Bayati[1]

[1]PEAK Wind, Jens Baggesensvej 90K st, 8200 Aarhus N, Denmark
[2]DTU, Frederiksborgvej 399, 115, S20, 4000 Roskilde, Denmark

**Correspondence:** Azélice Ludot (ajmlu@dtu.dk)

**Abstract.**

As the first commercial floating wind projects are about to enter the market, numerous operational challenges are still to be addressed. One significant challenge is the Operations and Maintenance (O&M) cost, which currently represents 30% of the overall cost of energy. While reducing this cost is essential for all wind turbines, it is especially critical for floating wind due to the increased complexity and logistical challenges of maintaining turbines in deeper, more remote offshore locations. Consequently, a significant part of recent developments in the wind industry focus on the condition monitoring of mooring systems, which are crucial for the structure's integrity. Failures in mooring systems are often due to fatigue damage, which can be monitored using real-time methods to improve maintenance predictions and repair planning. However, the high cost of the sensors, along with their reliability issues, such as frequent recalibration needs, can prevent their adoption at a commercial scale. Thus, this paper presents a methodology to develop machine learning-based surrogate models designed to predict, in real-time, hourly fatigue damage accumulation in the catenary chain mooring lines of floating wind turbines. This prediction is based on hourly measurements of five metocean variables: wind speed, wind direction, wave height, wave period, and wind-wave misalignment. Typically, the literature only accounts for the first three variables, assuming co-linear wave and wind, which can lead to unrealistic failure likelihood predictions for mooring systems. Additionally, the effects of corrosion and mean loads on chain fatigue damage are also considered, as they significantly affect the chain fatigue lifetime. The proposed tool is intended for predictive maintenance applications, which has been identified as a key area for cost reduction in floating wind, and can also be applied to reliability assessment purposes. In this paper, we describe the construction of a site-agnostic synthetic database of metocean conditions and corresponding fatigue damage values. This database is designed to be computationally efficient while also extensive enough to ensure that the surrogate model achieves strong performance. The advantage of using a site-agnostic metocean database is that it enables a simpler implementation while still accounting for the interdependencies of metocean variables, unlike site-specific databases that require deriving joint metocean distributions It also enables testing the fatigue of the same technology developed for different sites. To evaluate the tool's potential for existing projects, we quantified the uncertainties arising from both the model approximation and the statistical nature of the inputs. The methodology is applied to the mooring lines of the IEA 15MW UMaine semi-submersible. Training data is generated through the post-processing of tension time-series, extracted from OpenFAST simulations. Results demonstrated that hourly fatigue damage can be predicted using a gradient-boosted decision tree surrogate model, achieving an $R^2$ value of 0.928 with the defined





tuning strategy. This limitation in the $R^2$-value is a consequence of the seed-to-seed uncertainty and this can be reduced further with the integration of more realizations. Indeed, from the uncertainty quantification, it was found that the phase resolution is driving the accuracy of the model. The prediction is completed in less than 0.01 seconds, making it suitable for real-time asset

monitoring. The model's strong capabilities in terms of prediction speed makes it also particularly well-suited for surrogate-assisted reliability-based design optimization (RBDO). Unlike deterministic design optimization (DDO), RBDO incorporates design uncertainties by including the reliability index as part of the objective function. Using such surrogate models drastically reduces the computational cost associated with evaluating fatigue reliability during each iteration of the optimization process. Additionally, this approach effectively accounts for the most critical environmental variables influencing fatigue, ensuring a

comprehensive and realistic assessment of structural performance under uncertain conditions.

## 1 Introduction

In order to achieve the hoped-for carbon neutrality by 2050, the European Union has pledged to support the development of the offshore wind industry. The objective is to ramp up Europe's offshore wind generation capacity, aiming for a minimum of 60 GW by 2030 and 300 GW by 2050 (European Commission (2020)). This would constitute 25% of its electricity pro-

duction. Compared to fixed solutions, floating wind turbines offer a promising solution by enabling projects to be deployed in deeper waters, farther from the shore, and in areas with stronger winds, significantly expanding the potential sites for turbine installation.

    However, several obstacles still need to be addressed to fully capitalize on these opportunities. Mooring systems play a critical role in the integrity of floating offshore wind assets. Traditionally, the assessment of reliability and failure of mooring

systems for Floating Offshore Wind Turbines (FOWTs) relies on data and methods from the Oil & Gas (O&G) sector, where historical records indicate relatively high rates of mooring failures. Though, unlike traditional floating marine structures from O&G, the mooring systems of FOWTs endure increased and fluctuating loads due to the turbine's dynamic effects. These, along with additional loads from the growing sizes of turbines, contribute to significant uncertainties regarding project risks. To mitigate the risk of potential mooring line failures, designers typically opt to incorporate higher levels of redundancy or

conservatism in mooring system design, which can lead to increased Capital Expenditure (CAPEX). However, introducing such a high level of conservatism could bring challenges for commercial-scale floating wind projects, as it could make them economically unviable. In order to reduce this conservatism, an interesting solution to mitigate the risk and prevent a failure is the real-time monitoring of the mooring lines condition. While some events or accidents are hard to prevent and anticipate, degradation mechanisms such as fatigue damage in the chains (which is the most prevalent cause of failure in chain mooring

systems during operations (Fontaine et al. (2014))) can be tracked and quantified. This is usually done through the processing of mooring line tensions, which are measured or derived from on-site sensors. However, the cost of tension sensors and their low reliability, and the computational cost of fatigue calculations (if they have to be done for a large database throughout the turbine lifetime), challenges the application of this strategy. To address this problem, an important number of papers have been proposing solutions to both issues through the use of surrogate models. A surrogate model is a method used when the system





of interest cannot be easily modelled or computed, then an approximate mathematical model of the outcome is used instead. Focusing on papers related to offshore wind, Jayasinghe et al. (2023), Walker et al. (2022) and Angulo et al. (2017) developed such models to predict tensions based on the platform motions and metocean conditions. When it comes to accelerating the computation of the damage lifetime, several approaches have been developed (Müller et al. (2017), Teixeira et al. (2017) and Dimitrov et al. (2018)) combining the use of surrogate models to predict equivalent loads and then response surface or Kringing methods to derive the lifetime of the asset. All previous papers agreed on the potential of using surrogate models to derive damage lifetime. However, there is limited literature on the application of these models for condition monitoring and digital twin technologies. Additionally, in the recent publication tackling reliability analyisis of mooring lines (Hallowell et al. (2018), Zhao et al. (2023) and Safari et al. (2024)), authors agree on the influence of wind and wave directions on short and long term fatigue damage, whereas they are often assumed to be co-linear, which can lead to unrealistic failure likehood predictions for mooring systems.

In this paper, we present a methodology for developing a surrogate model that accurately predicts real-time fatigue damage accumulation in the mooring lines of a floating offshore wind turbine (FOWT), based on on-site measurements of five governing metocean variables: significant wave height, wave peak period, mean wind speed, wind direction and wind-wave misalignment. The proposed methodology aims to design a training database that ensures the accuracy of the surrogate model while minimizing computational costs, implement a tuning and training strategy to prevent overfitting and ensure robust model performance, and quantify the uncertainties introduced by both the model and the training data composition. This methodology is then applied to a straightforward example using the open-source IEA 15MW turbine with the UMaine VolturnUS semi-submersible floater and catenary chain mooring lines. The positive results obtained demonstrate the potential of using robust surrogate models for real-time asset monitoring as well as for reliability assessment problems. The novelty of this work lies in the consideration of five metocean variables, instead of the usual three, which often neglect the influence of wind and wave directions. Additionally, the model is site-agnostic, making it adaptable to different locations, and particular attention is given to the uncertainties introduced by the construction of the training database.

## 2 Reference System

This work examines the International Energy Agency (IEA) 15-megawatt reference offshore wind turbine, used with the University of Maine (UMaine) VolturnUS steel semi-submersible floater. The mooring system is composed of three chain catenary lines, each 850 meters in length. They are initially designed for the Gulf of Maine at 200 meters water depth. The properties of the turbine can be seen in Table 1, and the properties of the floater in Table 2.





**Table 1.** Key parameters for the IEA Wind 15-MW Turbine (Gaertner et al. (2020)). The rated wind speed is given at hub height.

| Parameter | Unit | Value |
|---|---|---|
| Power Rating | MW | 15 |
| Turbine Class | - | IEC Class 1B |
| Rated Wind Speed | m/s | 10.59 |
| Rotor Diameter | m | 240 |
| Hub Height | m | 150 |

**Table 2.** UMaine floater and mooring systems properties (Allen et al. (2020)).

| Parameter | Unit | Value |
|---|---|---|
| Draft | m | 20 |
| Platform Mass | t | 17839 |
| Mooring system type | - | Chain Catenary |
| Line Type | - | R3 Studless Mooring Chain |
| Number of lines | - | 3 |
| Anchor depth | m | 200 |
| Nominal Chain Diameter | mm | 185 |

In order to minimize the loads on the mooring lines, the system is aligned so that the mooring line 1 is facing the main incoming wave direction of the site.

## 3 Generation of the synthetic database

As presented in the introduction, this work aims to provide a methodology for developing a surrogate model that predicts accumulated fatigue damage over one hour periods based on a given set of statistical metocean variables. The first challenge lies in defining the appropriate inputs and corresponding outputs needed to construct the database. The loads on the mooring lines are primarily governed by the motions of the floater, which are influenced by hydrodynamic wave loading and aerodynamic forces on the turbine and tower. These wave and wind conditions are characterized by the following metocean variables:

- $U_{10}$: wind speed measured at a height of 10 meters. The wind speed at hub height will be extrapolated from the power law profile to be used as input for the time-domain simulations (International Electrotechnical Commission (2019)).

- $H_s$: significant wave height,





- $T_p$: wave peak period,

- $\theta_{\mathrm{wind}}$: wind direction,

- $\theta_{\mathrm{mis}}$: wind-wave misalignment angle.

These variables are typically provided by on-site metocean measurement buoys on an hourly basis. Since the inputs are statistical properties rather than time-series data, the output of interest in this paper is the prediction of accumulated fatigue damage in the mooring line sections corresponding to the given metocean conditions.

Figure 1 illustrates the overall methodology used to achieve the paper's objectives. The left column describes the model's training process, which includes database generation and model training. The right column describes the deployment of the model on-site once it is tuned and trained. This has not been addressed in this work.

The process described in subsection 3.1 represents the first step of the training workflow outlined in Figure 1. This step focuses on creating an efficient Design of Experiments that samples the input space more effectively, requiring fewer samples compared to methods like grid-based sampling. The next step, outlined in subsection 3.2, involves calculating synthetic damage values for each defined sample point through the post-processing of high-fidelity time-domain simulations. Finally, section 4 refers to the last step in the left column, where the surrogate model's hyperparameters are defined and the model training is conducted.

## 3.1 Variable space definition and Experimental Design

The challenge in defining the space of input variables lies in its significant influence on the quality of the surrogate model's training and performances. The aim is to avoid training the model for conditions it might never encounter, while ensuring it performs well under the most likely conditions. Therefore, two concurrent goals are faced: creating the most extensive training database that our computational capabilities allow to ensure the model performs well across the entire design space, while reducing the number of training points to save time and increase efficiency. This challenge has been addressed in the literature through sampling and Design of Experiment (DoE) theories (Leimeister and Kolios (2018)). DoE is an approach used in scientific research and industry to plan, conduct, and analyze experiments efficiently and effectively. The goal of DoE is to obtain reliable and statistically valid results while minimizing the number of experiments required and controlling for sources of variability. Then, this section will cover the definition of the input space, the sampling method used and how correlation between variables is accounted for.

### 3.1.1 Defining the range of input variables

The choice of ranges for variation in the input variables needs to balance two objectives: covering as much as possible potential sites of deployment, and ensuring that the selected samples are physically meaningful. Indeed to define input ranges of surrogate models' training database, two approaches, discussed by Dimitrov et al. (2018), can be followed:

- Site-specific training: Data points are selected from a specific site, so the model will only be able to predict damage for that particular location.





**Figure 1.** Overview of the methodology used (graph inspired by Dimitrov et al. (2018)).



– Non-site-specific training: Training data ranges are gathered from a variety of sites across a broader region, such as the North Atlantic. This approach allows the model to be applied to multiple locations within the entire region, rather than being specific to a single site.

In this study the second approach was chosen, to enable a wide deployment of the final model. Thus, metocean data over 25 years (which is the typical wind farm lifetime) is gathered from different sites in a given region. Then, lower and upper bounds are defined for each variable (which can be conditional on other variables, depending on the correlation between them) so that between 99.7%-99.9% of the data are included in the bounds. This ensures properly capturing the majority of conditions driving fatigue accumulation, as well as capturing a range of extreme conditions without putting too much emphasis on extremely rare events.

### 3.1.2 Sampling procedure

Once these conditional boundary boxes have been defined, points must be strategically selected within this space to gather data for building an accurate and representative model. A key requirement is to maintain the correlation between different variables during this selection process, in order to constrain the variables ranges. Then, these two main objectives in this sampling procedure can be summarized as:

– Select points to efficiently and representatively fill the space, with more points in the most probable areas and fewer in the extreme zones.

– Select points so that a chosen set of $(U_{10}, H_s, T_p, \theta_{wind}, \theta_{mis})$ is realistic and preserves the correlation between the variables.

Generating correlated samples that account for this dependence can be achieved using transformations like Nataf or Rosenblatt, which are commonly used to sample from multiple correlated random variables. In this work, a Rosenblatt transformation is applied, assuming wind speed is independent of other variables and follows a Weibull distribution, while the other variables follow uniform distributions within their defined conditional bounds.

To achieve the first goal of efficient point selection, the method for sampling from the defined distributions is crucial. Various sampling methods are used in the literature. For example, Murcia et al. (2018) use a quasi Monte-Carlo (MC) approach with a low-discrepancy sequence. Quasi-random Monte Carlo sampling, is a method used to generate sample points in a multi-dimensional space. Compared to traditional Monte Carlo methods that use purely random sampling, quasi-random methods aim to distribute sample points more uniformly across the space. This leads to faster convergence and more accurate estimates in numerical integration for instance. Müller et al. (2017) use Latin Hypercube Sampling to train an artificial neural network model for deriving damage equivalent loads for tower bending moments and fairlead tension. Tang et al. (2023) use uncertainty sampling for metocean data. Latin Hypercube Sampling and uncertainty sampling are also two methods that ensure a more uniform sampling than a complete random sampling.

In the present study, the space filling method that will be used is the quasi Monte-Carlo sampling. This method has been chosen from its very low discrepancy (when evaluated to random sampling (Dimitrov et al. (2018))) and its simplicity to under-





stand. The low-discrepancy sequence used is the Halton sequence (Faure and Lemieux (2009)), widely used in the literature,
partly due to its intuitive nature and ease of implementation. The Halton sequence is applied by sequentially taking all points
in the quasi-random series without omission or repetition.

## 3.2 High-fidelity simulations and damage computation

Fatigue damage of chains is derived from the tension history. In the absence of available data from operational floating wind
units, primarily due to a scarcity of open-source information, a history of time-domain tensions over various meteocean con-
ditions has been generated using software simulations. Among the suite of aero-hydro-servo-elastic softwares, OpenFAST has
been selected for this project because of its provision of open-source models for existing turbines and floaters, and its important
user community. OpenFAST (OpenFAST Developers (2023)) is an open-source software package developed at the National
Renewable Energy Lab (NREL).

### 3.2.1 Transient time

Transients effects and run-in-time have been investigated by Müller et al. (2018) for the LIFES50+ project. Summarizing the
results, for floating wind turbines under DLC 1.2 conditions, transients are expected to be between $500 - 1000\,\mathrm{s}$, provided cor-
rect initial conditions[1] regarding wind turbine performance, and with or without correct initial conditions regarding positioning
and displacements. In this project, having pre-computed the initial conditions, the transient time is then chosen to be equal to
$600\,\mathrm{s}$, in order to agree with the bounds. The initial conditions of the following DOFs are considered: rotor speed, blade pitch
angle, and platform surge, sway, heave, roll, pitch and heave positions.

### 3.2.2 Simulation time length

From DNV-ST-0119 (Det Norske Veritas (2021b)), in order to adequately capture the effects associated with the natural fre-
quencies of floating support structures when loads and responses are to be determined, a sufficient length of the involved
simulations must be ensured. Then, a minimum of 3 hours simulation is recommended to adequately capture effects such as
nonlinearities, second order effects, and slowly varying responses, and to properly establish the design load effects on moor-
ing lines. These recommendations are confirmed by Müller et al. (2018). These standards and reports (DNV-ST-0119 and
LIFES50+) recommend as well the use of 6 seeds per realization for wind and waves generation , in order to account the phase
randomness. A seed in wind and wave simulations is a random number used to generate different realizations of turbulent wind
or wave conditions, allowing for variability in environmental inputs while maintaining the same overall statistical characteris-
tics. However, in the literature, the dominant trend is to run for 1 hour of simulation, which is seen as a compromise between
capturing the nonlinearities and managing computational time. This approach has been followed by Jayasinghe et al. (2023),
Li et al. (2018), Zhao et al. (2021) and Cevasco et al. (2018). In this paper, the same compromise will be made regarding the

---

[1]Initial conditions are defined as the initial positions or conditions of the system depending on the mean wind speed or other environmental conditions
whose mean value is different from 0.





simulated time, while still maintaining six realizations per metocean point to represent the phase variation of the loads. Table 3 summarizes these time settings.

**Table 3.** Simulation time settings

| Parameter | Unit | Value |
| --- | --- | --- |
| Simulation time length | s | 4200 |
| Transient time | s | 600 |
| Number of seeds | - | 6 |

### 3.2.3 Damage computation workflow

Fatigue failure, caused by cyclic stress, is a common occurrence in materials. This type of failure typically begins with the formation of micro-cracks due to stress concentration effects at surface irregularities (Adedipe et al. (2017)). Fatigue assessment of offshore mooring systems is required by relevant rules and standards (Det Norske Veritas (2021a)), to demonstrate a satisfactory level of resistance under exposure to cyclic loads. It is typically recommended that the fatigue analysis should be based on S-N curves and the linear damage hypothesis[2]. S-N curves illustrate the relationship between cyclic stress (S) and the number of cycles until failure (N) for a given material (Kolios et al. (2019)). These curves are typically generated through laboratory testing, where samples are subjected to constant amplitudes until failure occurs. Recently, DNV introduced an updated version of these curves for use in the fatigue design of offshore steel structures Det Norske Veritas (2024).

Typically, an S-N curve is expressed as:

$$N = a \cdot S^{-m} \tag{1}$$

Where:

- $S$ in the given cyclic stress.

- $N$ in the number of cycle until failure corresponding to the cyclic stress $S$.

- $a$ in the intercept parameter and $m$ in the slope parameter. Both depends on the material considered.

These curves are based on fatigue tests of new chains, considering a fixed value of mean load of 20% of the minimum breaking load (MBL). Then, in the following fatigue calculations, the actual value of the mean load in the considered environment is neglected. However, Gabrielsen et al. (2019) showed that an increasing mean load could significantly reduce the fatigue life time of a mooring line. There is then an important interest in taking this effect into account when assessing the fatigue.

From Fontaine et al. (2014) and Scheu et al. (2019), it was seen that the corrosion phenomena accounts for an important part of the observed failures. Indeed, the crack propagation can be accelerated by the corrosion which concentrates stresses (pitting

---

[2]Under this hypothesis, the different contributions of the damaging cycles can be summed up to get the total damage value over the considered time.





effect). In the current standards, corrosion models are simply based on a reduction of the chain diameter using a constant rate per year. However, these simple models disregard the pitting effect, which was shown to have an important effect on the fatigue lifetime of the mooring lines (Gabrielsen et al. (2019)), more than the standards are actually predicting.

Hence, Lone et al. (2021) proposed an extended S-N curve formulation to include mean load and corrosion effects, by
expressing the intercept parameter of the S-N curve model as a function of the mean load and a corrosion grade indicator. They accessed the impact of including realistic corrosion levels compared to the designs codes, and highlighted the importance of take them into account in order to avoid non-conservative fatigue damage estimates. The updated formulation is the following:

$$\log N = b_0 + b_1 \cdot g_1(\sigma_m) + b_2 \cdot g_2(c) - m \cdot \log S \qquad (2)$$

Where:

- $(b_j)_{j \in \{0,1,2\}}$ are empirical coefficients, estimated from full scale fatigue tests of both used and new chains, tested under various conditions.

- $g_1(\sigma_m)$ and $g_2(c)$ are functions of the mean stress $\sigma_m$ and a corrosion grade $c$ respectively. The corrosion grade is defined according to a custom scale from 1 (new chain or mild corrosion) to 7 (severe corrosion), from Lone et al. (2021).

Based on full scale fatigue test data for used and new chains used in Lone et al. (2021) the following extended S-N design curve formulations have been found to provide the best and most reasonable fits to the data set:

$$\log N = 11.904 - 0.0507 \cdot \lambda_m - 0.106 \cdot c - 3.0 \cdot \log S \qquad (3)$$

Which $\lambda_m$ the mean load ratio, expressed in % of the MBL.

As the environmental loading are varying in amplitude with time, the cyclic stress amplitude is not constant. In order to
account for the fatigue effect of each stress cycle, one method to compute the resulting damage over a given period of time is to use the Palmgren-Miner hypothesis on linear accumulation:

$$D = \sum_i \frac{n_i}{N(\sigma_{m,i}, c_i)} \qquad (4)$$

Where D is then the fatigue damage and $n_i$ the number of cycles with the stress range $s_i$, mean stress $\sigma_{m,i}$ and corrosion grade $c_i$, and $N$ the corresponding number of cycles until failures under these conditions, derived from Equation 3.
In this work, the chosen workflow to compute the short-term fatigue damage is the time-domain approach based on a counting procedure (rainflow counting in this study) and a cumulative damage rule (Palmgren-Miner's rule) from stress time histories. Rainflow counting is a method to determine the number of fatigue cycles present in a load-time history.

### 3.3 Structure of the training database

From the described sampling procedure above, $n_{samples}$ sets of $(U_{10}, H_s, T_p, \theta_{\text{wind}}, \theta_{\text{mis}})$ will be sampled. Due to the computa-
tional cost of OpenFAST simulations and the required number of seeds, the number of metocean sampled points is limited to



| | $U_{10}$ | $H_s$ | $T_p$ | $\theta_{wind}$ | $\theta_{mis}$ | $\overline{D_{1h}}$ |
|---|---|---|---|---|---|---|
| $n_{samples}$ = 1000 | ... | ... | ... | ... | ... | ... |
| | ... | ... | ... | ... | ... | ... |

**Figure 2.** Structure of the built database

1000. However, to satisfy the requirements in terms of seeds, 6 simulations will be performed per sampled points (6 different seeds for wind and wave generation for each sampled point) resulting in the 6 damage values per point. To mitigate the influence of the phase of the loads on the damage values, the mean values of the damage over these six realizations will be used for the training database. Figure 2 illustrates the structure of the resulting database.

## 4 Surrogate models: definition and training strategy

The methodology described above for computing fatigue damage from the tension time series is time-consuming, involving OpenFAST simulations and rainflow counting. Therefore, there is a significant interest in speeding up the process, moving directly from the inputs (such as metocean conditions) to the resulting damage value. This can be achieved through the use of functions that are mapping the behavior of the system, called surrogate models. Once these models are trained on a dataset, damage can be predicted over time. The advantage of this method is that the computationally expensive simulations are only performed once during the training phase. During model deployment, only the tuned hyperparameters are used as inputs, and the prediction of the damage accumulation over one hour can be generated in less than one second, compared to 40 minutes using the complete workflow of OpenFAST simulations and damage computation. In this section, five surrogate model aletrnatives will be trained and compared to identify the best performer for the studied case. The first subsection will introduce the chosen models. The second subsection will present the tuning and training strategies. The third subsection will detail the evaluation of the models' performances to select the best one. Finally, the last subsection will discuss how to assess the uncertainties of the selected model.

### 4.1 Models definition

Five different regression models have been chosen for comparison, ranging from the simplest to the more complex: Gaussian Process Regression (Wilkie and Galasso (2021)), Support Vector Regression, Random Forest Regression (James et al. (2023)), LightGBM (Ju et al. (2019)), and XGBoost (Trizoglou et al. (2021)). The last two models are gradient-boosted decision tree models. In the choice of the model alternatives, tree-based models have been prefereed to neural networks as they have a better explainability and potentially better computational time (Zhang and Dimitrov (2024)). In supervised machine learning, models learn from training data to predict outcomes on unseen datasets, but they need to be tuned using hyperparameters. Properly tuning these hyperparameters is crucial, as they can significantly affect both the training process and the model's performance





on unseen data. Table 4 summarizes all the main characteristics, advantages and disadvantages of the five models described above.

**Table 4.** Main advantages and disadvantages of the five selected surrogate models.

| Method | Main Characteristics | Advantages | Disavantages |
|---|---|---|---|
| Gaussian Process Regression | Probabilistic prediction. Uses kernel to handle the non-linearity of data. | Confidence intervals can be easily computed from the probabilistic prediction. | Lose efficiency with increasing amount of data. |
| Support Vector Regression (SVR) | Uses kernel trick to handle non-linear data. Supports various kernels (linear, polynomial, RBF, sigmoid). | Effective in high-dimensional spaces. Robust to overfitting with the right kernel. | Requires careful tuning of hyperparameters. Computationally expensive for large datasets. |
| Random Forest Regression | Ensemble method using multiple decision trees. Each tree is trained on a random subset of the data. | Handles non-linear data well. Reduces overfitting by averaging multiple trees. | Can be less interpretable than single decision trees. Requires more memory and computational power. |
| Gradient Boosting Regression (XGBoost) | Sequential ensemble method that builds trees in a stage-wise fashion to correct errors of previous trees. | High performance and accuracy. Handles non-linear data well. Robust to overfitting with proper tuning. | Can be computationally expensive. Requires careful tuning of hyperparameters |
| LightGBM | Gradient boosting framework that uses tree-based learning algorithms. Focuses on speed and efficiency | Fast training speed and high efficiency. Handles large datasets and high-dimensional data well. | Sensitive to overfitting if not properly tuned. Requires categorical features to be preprocessed. |

## 4.2 Training strategy

Using the same data to both derive the parameters of a prediction function and test its performance is a methodological malpractice. This approach could result in a model that perfectly reproduces the known values but fails to generalize to new, unseen data, a phenomenon known as overfitting. To avoid this issue, it is common practice in supervised machine learning to set aside a portion of the available data as a test set. Typically, 70%-80% of the database is used for training and validation, and the remaining 30%-20% of the database is used for the test. The points that constitute this portion are chosen randomly.

When evaluating different sets of hyperparameters for models, there is still a risk of overfitting on the test set, as the hyperparameters will be tuned until the model reaches its optimal performance. Then, the model's capability to generalize will



be significantly reduced. To address this issue, another portion of the dataset can be reserved as a validation set. Initially, the model is trained on the training set, and then its performance is assessed on the validation set. After the experiment demonstrates success on the validation set, the final evaluation is be conducted on the test set.

However, by partitioning the initial database in three, its size is drastically reduced and the resulting hyperparameters might be very dependent on the chosen training, test and validation sets. To avoid again this, a solution is to perform a cross-validation (CV), where a test set is always retained for the final evaluation but no validation set is required (James et al. (2023)). There are several cross-validation procedures, but the most classic one is the k-fold cross-validation: the training dataset is split into $k$ subsets. Then, the following procedure is followed for every fold $k$:

  – The model is trained using $k-1$ of the folds as the training data.

– The resulting model is validated on the remaining part (the $k$-th fold is used as a test set)

The performance metric resulting through $k$-fold cross-validation is the mean of the values calculated during each iteration of the process.

To evaluate all possible combinations of hyperparameters and find the optimal configuration, Bayesian optimization has become a popular method for hyperparameter tuning. This approach uses probabilistic models to predict the performance of

different hyperparameter sets, selecting the next set to evaluate based on past results to minimize the number of evaluations needed. Bayesian optimization is especially useful when evaluating each hyperparameter configuration is time-consuming. However, in this study, the number of hyperparameters per alternative model is limited, and evaluations are fast, so simpler methods like Grid Search and Random Search will be used. Grid Search systematically evaluates all possible combinations within specified ranges, while Random Search samples hyperparameters randomly from predefined distributions, offering a

more resource-efficient approach. In the case of several hyper-parameters to be tuned (for SVR, RF and gradient boosted decision trees for instance), this method will be preferred. The training strategy applied in this work is illustrated on Figure 3, and Figure 4 represents the cross-validation process.

## 4.3   Evaluate the performance of the models

After tuning the hyperparameters and training the resulting model, it undergoes testing on the test dataset, followed by perfor-

mance evaluation. Performance evaluation involves comparing predicted values $\hat{y}_i$ to actual values $y_i$ through the computation of residuals $\epsilon_i$ for each point:

$$\epsilon_i = \hat{y}_i - y_i \tag{5}$$

Analyzing the distribution of residuals, including their mean and variance, provides insights into prediction quality and identifies potential overfitting or underfitting. Further metrics enhance understanding of model performance. The R-squared

value ($R^2$), a widely used metric, measures the goodness of fit of a regression model, ranging from 0 to 1; a value of 1 indicating a perfect fit. The residuals analysis offer insights into the magnitude of differences between actual and predicted values. The $R^2$ is only sensitive to correlation between variables (between target values and predictions), but does not capture bias. Residuals




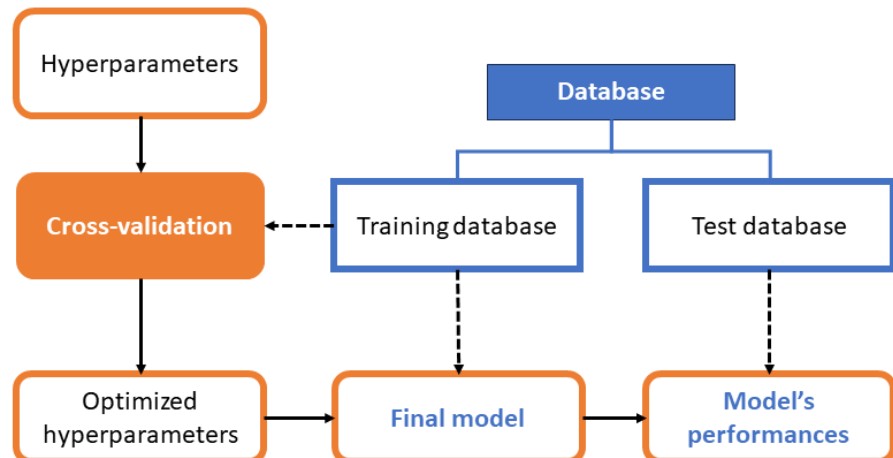

**Figure 3.** Flowchart of the cross-validation workflow performed in the model training phase.

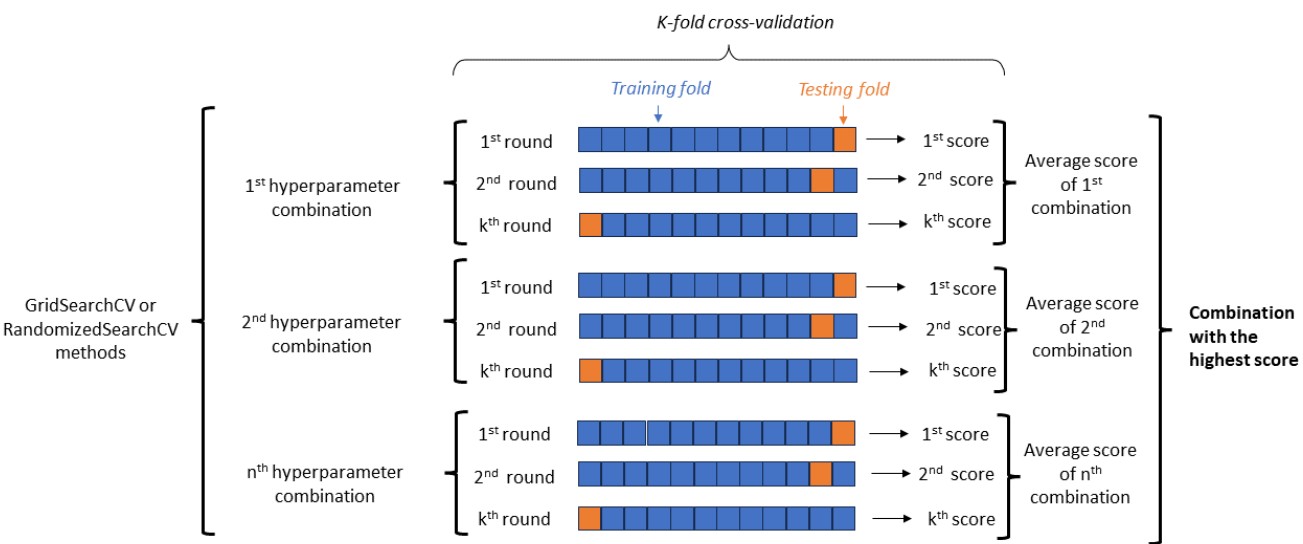

**Figure 4.** Illustration of the tuning strategy of the model's hyperparameters and performances evaluation




are sensitive to both correlation and bias, but can't distinguish between the two. The mean absolute pourcentage errors (MAPE) is sensitive to bias, but because it is in absolute terms, it also provides some information about the error variance. Hence, the use

of combination of R-squared and MAPE gives the possibility to both look at the correlation and the bias, and diagnose whether bad performance could be due to lack of correlation or high bias. However, these metrics lack information on error sign and variability. Therefore, studying error spread, sign, and variance is essential for comprehensive performance assessment. MAPE will be computed as follows:

$$MAPE = \frac{1}{n}\sum_{i=1}^{n}\frac{|\hat{y_i} - y_i|}{\hat{y_i}} \cdot 100 \tag{6}$$

**4.4 Uncertainties quantification**

To deploy the developed model effectively, it is crucial to quantify its uncertainties (Sullivan (2015)). Besides real-time monitoring, this model can also be used in probabilistic design approaches for reliability analysis. In such cases, due to the small target failure probabilities, the tails of the distributions, not just the bulk, become highly significant. The use of a surrogate model to approximate complex models and reduce computational time introduces additional uncertainties, amplifying any po-

tential inaccuracies of the surrogate model. In this context, it is assumed that the simulations perfectly represent real-world system, enabling to remain focus on uncertainties originating from the surrogate modeling process itself rather than discrepancies between simulations and reality.

Let's consider the inputs of our surrogate model, $\mathbf{x}$, and the output, $\mathbf{y}$. The true relationship between $\mathbf{x}$ and $\mathbf{y}$ is represented by a function $\tilde{g}(\mathbf{x})$, which is approximated by the surrogate function $g(\mathbf{x})$. $g(\mathbf{x})$ can be any of the five surrogate models described

earlier, and then does not achieve a perfect mapping of the true function. Thus, there will be an error $\epsilon_g$ when comparing the outcomes of $g(\mathbf{x})$ to $\tilde{g}(\mathbf{x})$:

$$\mathbf{y} = \tilde{g}(\mathbf{x}) = g(\mathbf{x},\theta) + \epsilon_g \tag{7}$$

Where $\theta$ represents a finite set of parameters of the function $g$. $\epsilon_g$, represents the so-called model error. It is an example of epistemic uncertainty, which is uncertainty that results from the approximation and that can be reduced by improving the

accuracy of the model. On top of model uncertainty, uncertainties can also arise from the inputs and outputs. In this study, as numerical simulations are used to create the database there are no measurements errors. However, from the stochastic nature (seed-to-seed variability) of OpenFAST, an aleatory uncertainty can arise from the output, which is called $\epsilon_y$. Then the true values can be expressed as: $y = \hat{y} + \epsilon_y$. With leads to the expression:

$$\hat{\mathbf{y}} + \epsilon_y = g(\hat{\mathbf{x}},\theta) + \epsilon_g \tag{8}$$

**4.4.1 Model uncertainty**

In order to give an estimate of $\epsilon_g$, an ensemble modelling approach will be used (Dimitrov et al. (2022)). It is a technique that combines multiple machine learning models to improve overall predictive performance. The basic idea is that a group





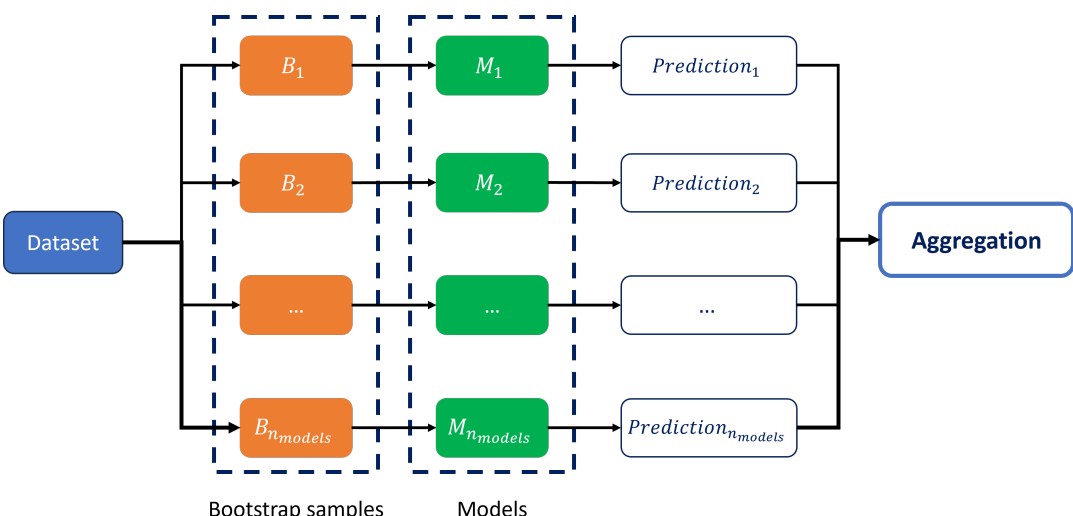

**Figure 5.** Illustration of the bagging process

of weak learners can come together to form one strong learner. An ensemble model typically consists of two steps: first multiple machine learning models are trained independently; then, their predictions are aggregated in some way, either by

voting, averaging, or weighting. This ensemble is then used to make the overall prediction. From these multiple predictions, the variance can as well be derived, which will provide an estimate of the model uncertainty. Here the ensemble model method used is the so-called bagging, which is an acronym for Bootstrap Aggregating (James et al. (2023)). Bagging is a technique in which a number of examples of the same base model are trained on distinct portions of the training data. The subsets are generated via bootstrapping, which is defined as a random sampling of the training data with replacement. Each base model is

trained on its own bootstrapped subset of data. Each of the models' predictions is then integrated using a voting or averaging procedure to produce the final prediction. Figure 5 illustrates this bagging procedure, where $n_{models}$ is the number of models that will be trained. The initial dataset is composed of $m_{total}$ points and the bootstrap samples are composed of $m_b$ points.

In this project, an ensemble modeling approach is employed to determine model uncertainties. Following the ensemble modeling theory defined above, the following steps need to be followed to derive the model uncertainties for this approach:

1. To isolate the effect of model uncertainty from aleatory uncertainties, the training data consists of the average output value $\bar{\mathbf{y}}$ of multiple realizations at the same input value combinations. The number of realizations is denoted $N_k$.

2. The database is split in a training and a test subset. The number of points in the test dataset is referred as $m_{test}$.

3. From the training subset, an ensemble of $n$ models are trained on $n_{models}$ different training subsets, generated through bootstrapping of the initial training database.

4. Predictions of the $n_{models}$ models are made on the test database.





5. The residuals of the $i^{th}$ model in predicting the output averaged over all the realizations from the input point $\mathbf{x}_j$ is denoted $\epsilon_{g,ij}$, and defined as:

$$\epsilon_{g,ij} = \bar{\mathbf{y}}_j - g_i(\mathbf{x}_{ij}, \theta_i) \quad \text{for } i = 1,\ldots,n_{models}, \quad \text{for } j = 1,\ldots,m_{test} \tag{9}$$

6. For each point $\mathbf{x}j$, the model uncertainty is represented by the standard deviation $\sigma_{\epsilon_{g,j}}$ of the residuals $\epsilon_{g,ij}$ across all indices $i = 1,\ldots,n_{models}$, for the given value of $j$.

### 4.4.2 Aleatory uncertainty of the output

The surrogate model used in this thesis was trained on averaged damage values across six realizations. This approach was chosen based on the assumption that averaging would reduce the variation in damage values for a given data point, which is caused by the phase randomness of the loads. To validate this approach, a comparison similar to that in Dimitrov et al. (2022) will be performed: the resulting aleatory uncertainty in the output, $\sigma_{\epsilon_{\bar{y}}}$, will be compared to the aleatory uncertainty from a model trained on a single realization per data point, $\sigma_{\epsilon_y}$.

$$\sigma_{\epsilon_y} = \sqrt{\frac{\sum_{k=1}^{N_k}(\mathbf{y}_k - \bar{\mathbf{y}})^2}{N_k}} \tag{10}$$

$$\sigma_{\epsilon_{\bar{y}}} = \frac{\sigma_{\epsilon_y}}{\sqrt{N_k}} \tag{11}$$

These resulting quantities will be described in the results section.

## 5 Environmental conditions

To design the input variable space for building the Design of Experiments, data spanning 25 years from various sites are collected to establish conditional bounds for the five selected metocean variables. Eight locations along the European coast of the North Atlantic are chosen based on existing or planned floating wind farm projects and summarized in Table 5. Additionally, a depth filter is applied, focusing only on sites with water depths around 100 meters to ensure consistency in wave behavior and similar oceanographic conditions. For this work, the metocean data have been taken from the hindcast model of ResourceCODE developed by Ifremer (Raillard et al. (2022)). ResourceCODE wave hindcast model is based on a high-resolution unstructured grid extending from the south of Spain to the Faroe Islands and from the western Irish continental shelf to the Baltic Sea.





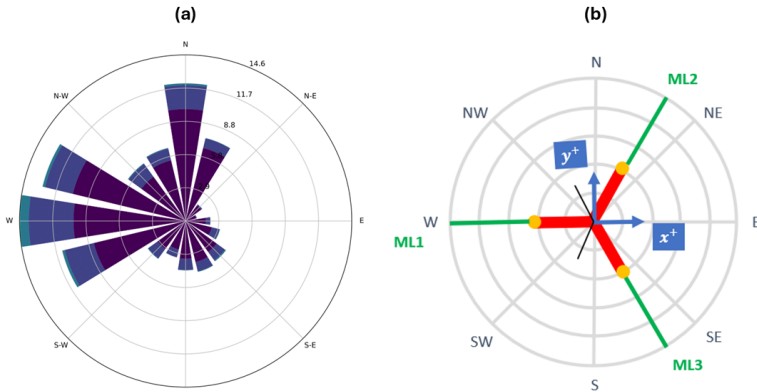

**Figure 6.** (a) Wave directions rose from the eight aggregated sites. (b) Layout of the reference mooring system.

**Table 5.** Description of selected sites

| Name | Location | Latitude [°] | Longitude [°] | Depth [m] |
|---|---|---|---|---|
| Projet Bretagne Sud AO5 | France | 47.25 | $-3.5$ | $75 - 100$ |
| WindFloat Atlantic | Portugal | 41.75 | $-9.25$ | 100 |
| Hywind Scotland | Scotland | 57.25 | 0.25 | $95 - 100$ |
| Erebus Floating Wind demo | England (Celtic Sea) | 51.50 | $-5.75$ | 70 |
| N2 | Scotland | 59.00 | $-5.50$ | 98 |
| MarramWind | Scotland | 58.25 | $-0.50$ | 111 |
| Arven (NE1) | Scotland | 60.25 | 0.00 | 100 |
| Ayre (NE2) | Scotland | 58.75 | $-2.25$ | 70 |

In order to minimize the loads on the mooring lines, the system is aligned so that mooring line 1 is facing the main incoming wave direction of the site. In the case studied here the prevailing wave direction from the eight considered sites is the West.

This approach of using variation ranges and conditional bounds makes the generated metocean database non-site-specific. To develop a site-specific database, however, the approach would need to use the joint distribution of metocean variables specific to the site. . For more than three variables, deriving these joint distribution functions requires significant effort. Parametric approach is the prevalent method employed in the literature to derive the joint probability of metocean variables. Parametric methods involve assuming specific probability distributions (e.g., Weibull for wind speed, Rayleigh for wave height) and

coupling them through copulas, which model the correlation between variables (Fazeres-Ferradosa et al. (2018), Li and Zhang (2020)). Another widely used approach is the conditional modeling method (Vanem et al. (2024)), where each variable (e.g., wave height) is modeled conditionally on others (e.g., wind speed). In the joint distribution developed by Vanem et al. (2024),



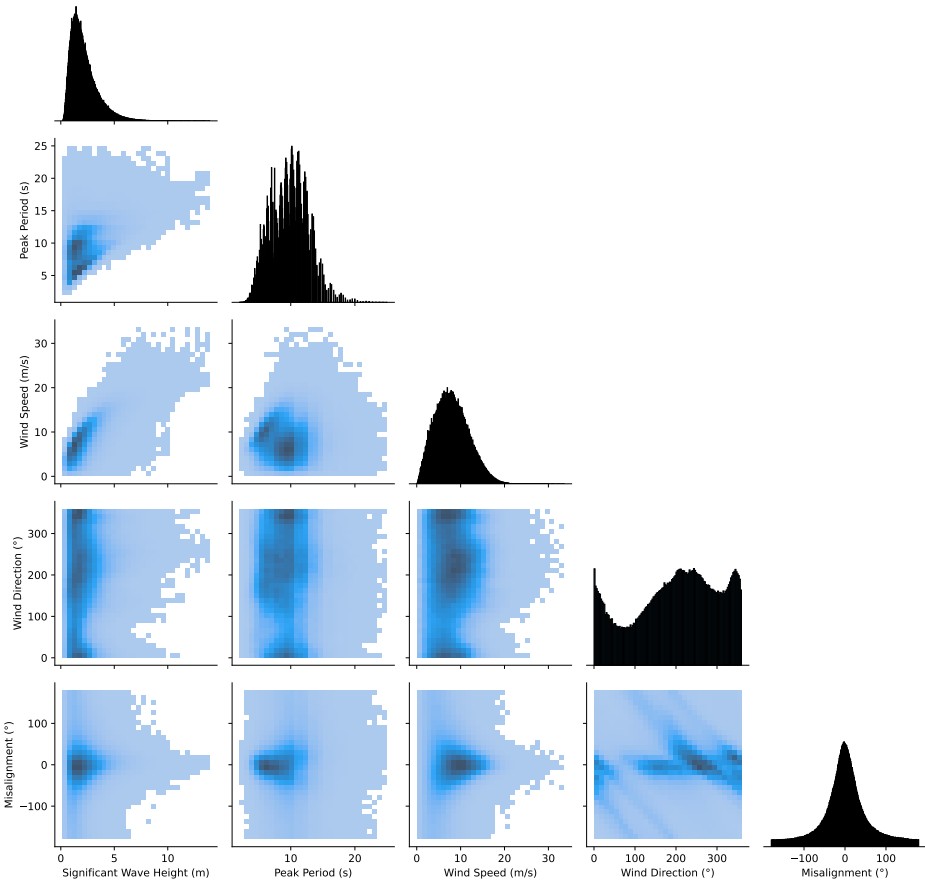

**Figure 7.** Pairwise scatter plots of wind and wave data (represented by blue dots, with darker shades indicating higher occurrence over the observed period). The distribution of each metocean variable is displayed in dark.

simplified conditional models are considered in their approach: wind direction, significant wave height, and wave direction are modeled as conditional only on wind speed, while peak wave period is modeled conditional on significant wave height. A

similar approach is followed in Det Norske Veritas (2021a). Accordingly, the same simplified models will be adopted in this study.

On Figure 7 the distributions and the bivariate plots of the extracted metocean from ResourceCODE variables over 25 years at the eight sites selected in Table 5 are displayed. Table 6 resumes the selected conditional bounds for each variable.




**Table 6.** Bounds of variation for the variables considered

| Variable | Lower bounds | Upper bounds | Distribution |
|---|---|---|---|
| $U_{10}$ | $0\,[\mathrm{m \cdot s^{-1}}]$ | $25\,[\mathrm{m \cdot s^{-1}}]$ | Weibull |
| $H_s$ | $0.132 + 0.009 \cdot U_{10}^2\,[\mathrm{m}]$ | $5.0 + 0.15 \cdot U_{10} + 0.004 \cdot U_{10}^2\,[\mathrm{m}]$ | Uniform |
| $T_p$ | $1.169275 \cdot H_s\,[\mathrm{s}]$ | $18.5\,[\mathrm{s}]$ | Uniform |
| $\theta_{wind}$ | for $U_{10} \in [0,15]\,[\mathrm{m}]$: $\quad -180\,[°]$<br>for $U_{10} \in [15,25]\,[\mathrm{m}]$: $U_{10}^{1.79802} - 326.227\,[°]$ | $180\,[°]$<br>$-U_{10}^{1.79802} + 326.227\,[°]$ | Uniform |
| $\theta_{mis}$ | for $U_{10} \in [0,15]\,[\mathrm{m}]$: $-180\,[°]$<br>for $U_{10} \in [15,25]\,[\mathrm{m}]$: $\frac{180}{\sqrt{25-15}}\sqrt{U_{10}-15} - 180\,[°]$ | $180\,[°]$<br>$-\frac{180}{\sqrt{25-15}}\sqrt{U_{10}-15} + 180\,[°]$ | Uniform |

## 6 Results

This section describes the results of the model selection, training, and uncertainty quantification. In this work, a surrogate model is specific to:

- A mooring line.
- A segment within this mooring line.
- A corrosion grade $c \in [\![1,7]\!]$.

Therefore, this section will present results only at the fairlead of the mooring line 1 (Figure 6), for the corrosion grade 3. The database used for these purposes is composed of 1000 sampled points, each with their corresponding damage value averaged over 6 realizations (6 different seeds for wind and wave generation for each sampled point).





In this entire section, the damage values are raised to the power of $1/m$ (where $m = 3$, the Wöhler coefficient from the S-N curves of the chains). Indeed, damage values can vary widely, and by taking the cubic root, this range of variation is

compressed, reducing variance and making the data more uniform. This transformation helps achieve better model training. Additionally, reducing the range of variation of the inputs can reduce the impact of potential outliers. Therefore, throughout this part, the damage values will be raised to this power.

## 6.1 Model selection and evaluation

Following the flowchart of Figure 3, the initial database is first split into a training set of 800 points and a testing set composed

of the remaining 200 points. For each set of hyperparameters to be tested, the model's performances will be assessed using k-fold cross-validation over the training set. The number of folds is set to 10. In the literature, the number of folds varies between 5 and 20. Choosing $k = 10$ balances the computational time required for performance evaluation over all folds and avoids having overly large folds in an already small dataset. These settings are summarizes in Table 7.

**Table 7.** Settings for the models' training and tuning

| Parameter | Value |
| --- | --- |
| Number of points in the dataset | 1000 |
| Number of seeds considered in the averaged damage value ($N_k$) | 6 |
| Size of the test dataset ($m_{test}$) | 200 |
| Number of folds ($n_{models}$) | 10 |

The five selected surrogate models are tuned according to the strategy defined earlier. The ranges of the hyperparameters to be

tested are first defined broad and then refined to reached the best estimation of the optimal set. The optimal $R^2$ values resulting from the tuning can be found in Table 9 and the corresponding optimal parameters are given in Table 8. The LightGBM and XGBoost models present the best results, with coefficients of determination over $0.9$. These $R^2$ values from the cross-validations are a good way to evaluate the accuracy performance of a model over any unseen dataset. Indeed, as explained in the methodology section, the coefficient of determination is calculated as the mean of all the $R^2$ values from the predictions

over all the folds of the k-fold cross-validation. This provides a good indication of which model will perform the best overall.





**Table 8.** Optimal hyperparameters of the five considered models after tuning.

| Model | Hyperparameter | Value |
|---|---|---|
| Gaussian Process Regression | Length Scale | 6.00 |
| Support Vector Regression | $\epsilon$ | 0.001 |
| | $C$ | 10 |
| Random Forest Regression | min_samples_leaf | 2 |
| | max_leaf_nodes | 100 |
| | max_features | 5 |
| XGBoost Regression | Learning rate | 0.081 |
| | n_estimators | 500 |
| LightGBM Regression | Learning rate | 0.125 |
| | max_depth | 8 |
| | num_leaves | 16 |

**Table 9.** Resulting $R^2$-values from the cross-validation over the training database for the five tested models.

| | $R^2$ **from the CV** |
|---|---|
| Gaussian Process Regression | 0.798 |
| Support Vector Regression | 0.858 |
| Random Forest Regression | 0.899 |
| XGBoost Regression | 0.916 |
| LightGBM Regression | 0.928 |

In addition of the $R^2$ coefficient, other quantities need to be evaluated to complement the information in order to perform a well-informed selection over the models. These tuned models are trained over the training database and deployed on the test set where their prediction is compared with the actual values of damage of the test dataset. Figure 8 presents these predictions of the five models and Figure 9 the residuals distributions. It can be observed that the best results are achieved with the most complex models (LightGBM and XGBoost), whereas the Gaussian Process and Support Vector models do not capture well the high non-linearity of the system. The three best performer models (RFR, LightGBM and XGBoost) perform well at low damage values, which correspond to mild environmental conditions that are more represented in the training dataset due to the




**Figure 8.** Plots comparing the damage values predicted by the models over the test dataset to the actual damage values from this test set.

sampling procedure. At higher damage values, the predicted values tend to diverge a bit more from the actual values, with even some very well marked outliers. Having a look on the residuals, they are centered on 0 as expected, with a significant smaller

standard deviation for the RFR, XGBoost and LightGBM than for the two first ones. XGBoost is slighlty shifted to the negative values. The outliers can be clearly seen on the fat-tail of the LightGBM residuals.



**Figure 9.** Distributions of residuals from models' predictions compared to actual mean damage values across the test dataset. Each plot includes the mean residual value and its standard deviation.



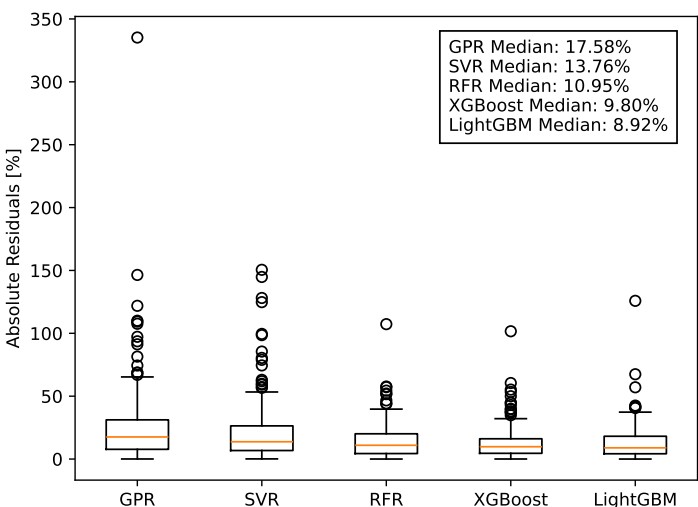

**Figure 10.** Absolute residuals for all tested models

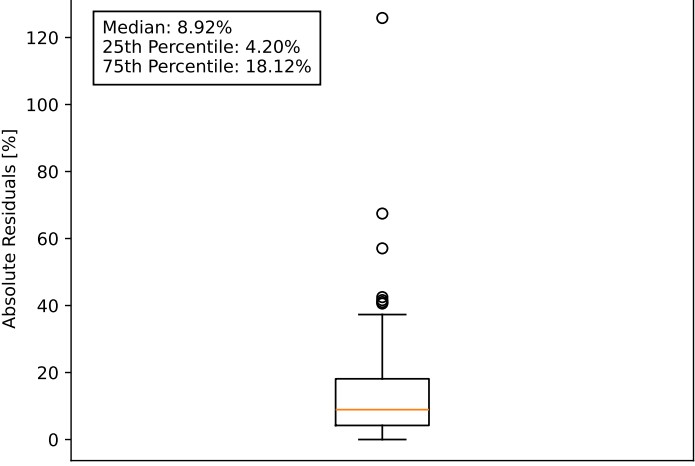

**Figure 11.** Absolute residuals for the LightGBM model





Figure 10 shows the accuracy of the predictions of the five considered models through the statistical analysis of the absolute residuals. Each box extends from the first quartile (Q25) to the third quartile (Q75) of the data, with a line at the median. The outlier values are plotted as circles. These statistical values are resumed in Table 11. LightGBM has the lowest median absolute
residual (8.92%), indicating it has the best performance among the models in terms of median error. The spread of absolute residuals (indicated by the box and whiskers) varies among the models, with GPR and SVR showing a wider spread compared to RFR, XGBoost, and LightGBM.

The selection of the model will primarily consider the $R^2$ value from cross-validation to gauge its potential for generalization. However, practical factors such as training and prediction times will also strongly influence the decision. Specifically,
preference will be given to models that offer high accuracy and low prediction times over those with the highest accuracy but longer prediction times. Results are detailed in Table 10, showing training and predicting times across a training database of 800 mean damage values from six seeds, and a test database of 200 points. The GPR and LightGBM models demonstrate significantly faster training times, approximately ten times quicker than other models. Additionally, the LightGBM model achieves predicting times around ten times faster than the GPR model.

**Table 10.** Performances of the five models over the testing database.

|  | Training Time [s] | Predicting Time [s] | RMSE | Standard Deviation |
|---|---|---|---|---|
| Gaussian Process Regression | 0.036 | 0.010 | 0.001084 | $1.168 \cdot 10^{-6}$ |
| Support Vector Regression | 0.378 | 0.017 | 0.000957 | $9.110 \cdot 10^{-7}$ |
| Random Forest Regression | 0.295 | 0.016 | 0.000696 | $4.836 \cdot 10^{-7}$ |
| XGBoost Regression | 0.353 | 0.003 | 0.000658 | $4.844 \cdot 10^{-7}$ |
| LightGBM Regression | 0.047 | 0.001 | 0.000668 | $4.435 \cdot 10^{-7}$ |

**Table 11.** Statistics over the absolute pourcentage residuals distribution for each model.

|  | Mean [%] | Median [%] | 25th percentile [%] | 75th percentile [%] |
|---|---|---|---|---|
| Gaussian Process Regression | 25.88 | 17.58 | 7.73 | 31.18 |
| Support Vector Regression | 21.59 | 13.76 | 6.77 | 26.39 |
| Random Forest Regression | 14.12 | 10.95 | 4.68 | 19.94 |
| XGBoost Regression | 12.79 | 9.80 | 4.56 | 16.12 |
| LightGBM Regression | 12.72 | 8.92 | 4.20 | 18.12 |

In the previous paragraphs, the models have been compared over their accuracy and their computational times. It results from this study that the model which matches the best coefficient of determination with the smaller computational time is the





**LightGBM model**. Therefore, this model will be used in the following sections, for the uncertainties quantification and model deployment.

## 6.2 Estimation of the uncertainties

To quantify both the model uncertainties of the tuned LightGBM model and the aleatory uncertainties arising from realization-to-realization variability, the methodology outlined in subsection 4.4 is applied.

### 6.2.1 Model uncertainties

The settings that are used to derive the model uncertainty in this work are summarized in Table 12.

**Table 12.** Settings of the ensemble modeling approach used to derive the model uncertainty

| Parameter | Value |
|---|---|
| Number of points in the dataset | 1000 |
| Number of realizations ($N_k$) | 6 |
| Size of the test dataset ($m_{test}$) | 200 |
| Number of models in the ensemble ($n_{models}$) | 15 |

Figure 12 illustrates the distribution of the model uncertainties, derived from a bagging analysis over 15 bootstrap samples
of the dataset. The model uncertainties are centered on zero and the tail is not significant. Moreover the order of magnitude of the model uncertainties is $10^{-8}$ which is very small (can be almost considered as 0). The uncertainties brought by the model not matching perfectly the system behavior are actually very small, and can be then considered are not significant. On top of that they are expected to reduce with a bigger dataset.

### 6.2.2 Aleatory uncertainties

Figure 13 depicts the seed-to-seed uncertainties alongside with the aleatory uncertainties from the database, after averaging the damage values over six realizations. The seed-to-seed uncertainties are computed by comparing tow-by-two the damage values resulting from time-domain simulations using two different seeds to generate the loads. They present a mean value of $6.834 \cdot 10^{-4}$. However, when using an averaged database, the aleatory part of the output uncertainty is reduced by a factor of $\sqrt{6} \approx 2.45$. This reduction can be seen on Figure 13. In this work, to limit the computational time, only six seeds were
considered to compute the averaged database. Thus, the more realizations will be accounted for in the mean damage calculated the most this value of uncertainty will be reduced. The mean aleatory uncertainties fall within the range of the calculated RMSE for the LightGBM model (Table 10). This indicates that the prediction errors observed earlier are primarily due to the variability introduced by seed-to-seed differences.



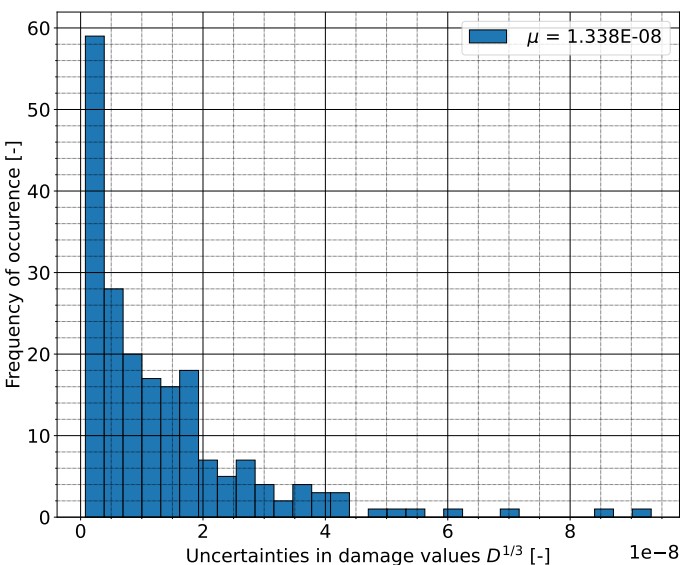

**Figure 12.** Distribution of the model uncertainties

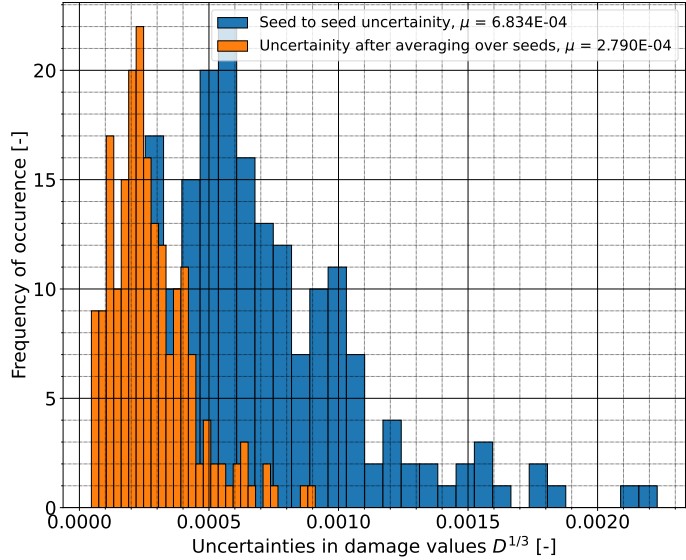

**Figure 13.** Distribution of the aleatory uncertainties





# 7 Discussion

This work presents a methodology for designing and generating a synthetic database of mooring line fatigue damage values based on corresponding five governing statistical metocean conditions. Emphasis is placed on the selection process of the surrogate model and the quantification of uncertainties introduced by this approach.

Five surrogate model alternatives were tested for this specific problem, with the best performers being tree-based models, known for their resistance to overfitting and minimal need for hyperparameter tuning. Among these, gradient-boosted decision

trees (GBDT), such as LightGBM and XGBoost, outperformed Random Forest models. GBDT builds trees sequentially, with each tree correcting the mistakes of the previous one, which reduces bias and improves accuracy. In contrast, Random Forest builds trees independently and averages the results, which can sometimes make them less accurate than a well-tuned GBDT. However, none of the models achieved an $R^2$-value exceeding 0.93, primarily due to seed-to-seed uncertainty in turbulence and wave seeds—a statistical variability that, over time, averages out to converge on long-term damage if the model is unbiased.

In other words, this $R^2$ limitation is due to short-term fluctuations caused by realization-to-realization uncertainty, though in the long run, performance depends more on model bias. Given this, it is unlikely that another surrogate model would significantly outperform using the same database. Improving the database would be necessary to reach greater accuracy. The very low model uncertainties compared to the aleatory uncertainties indicate that improving the training database - and thus the surrogate model's performance - depends more on using additional seeds to compute averaged damage values than on merely

increasing the sample size. However, LightGBM and XGBoost models still demonstrated good performances, on the limited database generated for the study. And on top that, they demonstrated great time performances, regarding the prediction time, which highlight their potential for real-time monitoring applications. Nevertheless, the median absolute residuals of LightGBM predictions remain high, raising concerns for its use in reliability assessment frameworks. Improvements are required before deployment. Comparing with models from the literature, Gradient Boosted Trees (GBDTs) have gained popularity especialy

when it comes to wind and wind power prediction, showing excellent accuracy. For instance, Sobolewski et al. (2023) reported MAE reductions from 8.20% to 3.84% for 48- to 248-hour wind power forecasts using meteorological data. Similarly, Park et al. (2023) achieved normalized MAE between 5% and 6% for wind power forecasting. GBDTs have seen limited application in loads assessment. Recently, Wang et al. (2025) used LightGBM to predict DELs for the NREL 5MW turbine, achieving an R-squared of 0.995 and mean absolute errors between 4% and 8% for tower loads. This suggest that there is potential

improvements of our model using a larger database, with a better inclusion of extreme values and performing some preparation work on the inputs.

Beyond surrogate model performances, a key novelty of this work is the integration of wind and wave directions as inputs for mooring line fatigue damage computation, whereas most studies typically assume aligned wind and waves for conservatism. While the conservative assumption of aligned wind and waves is well-documented, a global sensitivity analysis using tech-

niques like Sobol indices could help refine the surrogate model by identifying other key environmental and operational factors that influence mooring line fatigue, as the current and marine growth.





The surrogate model developed in this study is based on simulation results rather than real-world damage data, which introduces certain limitations. While this simulation-based approach demonstrates the feasibility of the method, its application to real-world scenarios requires further validation. The most robust approach involves using a database of measurements collected from operational systems. However, obtaining such measurements can be challenging due to data scarcity and practical limitations. In cases where simulation data must be used, it is essential to validate the model predictions using field measurements of fatigue damage obtained from structural health monitoring systems, such as strain gauges or load sensors installed on actual structures. This validation ensures that the simulation model accurately reflects the real system's behavior under varying environmental conditions. By comparing predicted and observed damage over time, the model's reliability and accuracy can be assessed. When measurement data is limited, insights from existing measurements can still enhance simulation-based models through techniques like physics-informed machine learning or transfer learning, as proposed by Schröder et al. (2022). These methods improve the performance of the model, bridging the gap between simulated and real-world scenarios, ensuring greater accuracy, and increasing applicability in practical settings.

The model's robustness could also be evaluated by applying it to multiple sites with varying metocean conditions and across different wind turbine and floater designs. Consistently accurate predictions across a range of systems would demonstrate its generalizability and reliability for real-world applications. Through these validation efforts and iterative improvements, the surrogate model has the potential to evolve from a simulation-based tool into a valuable component of real-world design optimization, reliability assessments, and operational monitoring frameworks.

A limitation of the methodology is its reliance on statistical data rather than phase-resolved data, which contributes to significant uncertainties. Moreover, the model has not been tested on unseen environmental conditions, and it primarily captures fatigue during normal power production, excluding scenarios like start-ups and shutdowns that may also contribute to fatigue.

An important characteristic of this work is the design-specific nature of the developed surrogate model, which depends on the wind turbine, floater, and mooring system design. If it were to be applied to a different floater or turbine, the surrogate model would need to be rebuilt using simulation data (or measurements, if available) for the new system. This limitation is not restrictive for reliability assessments or condition monitoring. With improvements in accuracy, the final surrogate model could indeed be used to assess the fatigue reliability of the VolturnUS mooring system at various sites across the North Atlantic, accounting for site-specific conditions. However, if this surrogate model were to be used in a design optimization framework with reliability as a constraint or objective, design variables would need to be included as inputs. Such models could significantly enhance Reliability-Based Design Optimization (RBDO) by reducing computational burden.

Then with further development, the outputs of this work could have multiple applications. The surrogate model could be used in digital twin applications for predictive maintenance, efficient repair scheduling, and managing component end-of-life, potentially reducing OPEX costs. Additionally, the fatigue reliability index of the mooring system could be derived at a reasonable computational cost, offering more site- and technology-specific failure rates, thus reducing the need for conservative risk margins in project planning. The site-agnostic approach offers several valuable use cases for early-stage project risk evaluation by specific floater design. This method allows for transferability without extensive re-parameterization, enabling cost-effective quantitative analysis for various applications. One key application is the evaluation of high-level CAPEX costs





and the selection of floater concepts, which can aid in choosing OEM preferred supplier agreements and potentially even enable commercial strategies such as framework agreements. From an OEM perspective (FOU + WTG), the site-agnostic approach aids parametric studies and scalability, allowing for site-agnostic decisions. This also touches on scalable Reliability-Based
Design Optimization (RBDO) as mentioned earlier. Additionally, the site-agnostic approach allows for the direct deployment of Digital Twins, incorporating a feedback learning loop from continuous sensor data. This enables the setup of predictive maintenance and reduces costs from the start, rather than waiting months or years into operations when data becomes available. This approach is cost-effective due to time savings and can be scaled across a wind farm fleet (i.e., positions) and wind farm portfolio (geographics).

## 8   Conclusion

The methodology developed in this paper demonstrated that fatigue damage in chain mooring lines can be predicted using a gradient-boosted decision tree surrogate model, achieving an $R^2$ value of 0.928 through a well-defined tuning strategy. With prediction times of less than $0.01\,\mathrm{s}$, the model is suitable for real-time asset monitoring and efficient fatigue reliability assessments. Two major sources of uncertainty were identified and quantified: model uncertainty, which reflects the model's ability
to map the system's behavior, and aleatory uncertainty, arising from variability in environmental loads across different seeds. Model uncertainty was found to be on the order of $10^{-8}$, underscoring the reliability of the surrogate model for deployment on assets. In contrast, aleatory uncertainty, which was roughly 10,000 times higher, emerged as the dominant contributor to overall uncertainty. Importantly, this uncertainty was shown to decrease with additional realizations, following a reduction factor of $\sqrt{n}$, where $n$ is the number of realizations used for averaging damage values.

This methodology would benefit from further research in several key areas. The model presented in this study relies on statistical quantities such as $H_s$ and $T_p$, whereas industry practices often utilize phase-resolved signals from turbine motions and time-series metocean data. Future work should explore the impact of excluding phase information from the load data. Additionally, as stated in the discussion, the model could be refined by identifying key environmental and operational factors that influence mooring line fatigue, through sensitivity analysis. Finally, extending this research to include reliability analysis
and failure rate assessments is crucial. By considering five metocean conditions instead of the traditional three, the model could reduce conservatism and provide more accurate failure estimates, alongside confidence intervals derived from quantified uncertainties.

The findings of this study provide key recommendations for the use of surrogate models in predictive maintenance. The uncertainty analysis highlights the importance of using a high number of realizations per input point to minimize aleatory
uncertainty, suggesting that prioritizing multiple realizations is more effective than simply expanding the dataset. Additionally, the time-efficient performance of the surrogate model makes it highly suitable for failure rate assessments and probabilistic design of floating systems.

In conclusion, this paper proposes a robust methodology for defining a database and selecting a surrogate model that incorporates multiple statistical variables to predict fatigue damage accurately. Its low computational cost shows great potential



for predictive maintenance applications and realistic failure rate assessments, offering a cost-effective approach to managing
        floating wind farms.





*Code and data availability.* The training dataset, along with the code for database generation and model training, is not publicly available. However the simulation dataset used in this study can be reproduced using publicly available tools (ResourceCODE, openFAST). Tuned values of the models' hyperparameters have been provided in the Results section.

*Author contributions.* A. Ludot proposed the methodology, created the database and developed the surrogate models presented in this article. A. Ludot also drafted the present paper. T.H. Snedker, A. Kolios and I. Bayati proposed, supported, discussed and reviewed the present work.

*Competing interests.* Some authors are members of the editorial board of WES journal.



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
