# Peer review of "Data-Driven Surrogate Models for Real-Time Fatigue Monitoring of Chain Mooring Lines in Floating Wind Turbines"

_Wind Energy Science, 2024_

## Referee Comment (RC2)

General comments:

The paper performs OpenFAST simulations of the 15-MW FOWT with a conventional mooring configuration under 1,000 sea states, and applies five surrogate models to evaluate their predictions of hourly mooring fatigue damage. The best surrogate model, which has the lowest $R^2$ values, is further used to estimate the uncertainty.

The paper is well-structured, and the topic of mooring fatigue monitoring is interesting, particularly with the use of surrogate models, which greatly improve efficiency. However, the novelty of the paper is not sufficiently demonstrated through the methods and results. The following comments are provided, with the hope of improving the quality of the paper.

1. Introduction part:
   a. In line 39, the paper mentions a target of 60 GW by 2030. Please verify this with the latest literature, such as the Global Wind Report 2024 by the Global Wind Energy Council, which sets a target of 320 GW by 2030.

   b. The introduction does not clearly demonstrate the novelty of the proposed surrogate model in this study. What new functions or methods does the proposed surrogate model introduce? Or is it merely incorporating two more environmental variables—wave period and wind-wave misalignment—into tension prediction for digital-twin technology? How importance of these two factors in mooring fatigue?
   In lines 45–60, the paper discusses the high risk of mooring failures in the offshore oil and gas (O&G) sector, and the mitigation of these risks using tension sensors for real-time measurement. Furthermore, the literature cited in lines 60–65 mentions a platform motion-based method that addresses the issues associated with tension sensors.
   However, the paper does not further elaborate on the novelty of the proposed approach. For instance, what is the specific importance of the surrogate model for condition monitoring? Why not use a GPS sensor directly instead of relying on a surrogate model? Since there is no interactive feedback for the operational or maintenance adjustments, but only post-processing of measurement data for fatigue prediction, how does the surrogate model or even digital twin technology offer a distinct advantage?

   c. The introduction lacks sufficient evidence to support the surrogate model's ability or digital twin technology to improve long-term mooring integrity in terms of fatigue. Fontaine et al. (2014), as the paper cited in line 55, found that 3 out of 29 mooring fatigue failures were caused by out-of-plane bending of chain links. In this case, in addition to tension values, the angles between two links are also crucial.

   However, the paper does not further discuss the capability of the platform motion-based surrogate model to predict the angles between two links. How did the surrogate models in literature (specially the platform motion-based method cited in line 60-65) or the proposed method for digital-twin technology, which in this paper incorporates two additional environmental variables, address this issue?

2. In the 'Reference system' section: please clarify the mooring pretension used in this study. Since varying pretensions influence the mooring stiffness and tension damage.

3. In the 'Generation of the synthetic database' section:
    a. the paper describes OpenFAST as a 'high-fidelity' tool; however, its official webpage (https://openfast.readthedocs.io/en/main/) refers to it as a 'multi-fidelity' tool. Typically, CFD simulations are classified as 'high-fidelity'.
    b. In line 116, the paper states that the sampling technique aims to avoid conditions that would never occur. Please clarify this sampling method further, especially in the context of using the 'non-site-specific training' approach for input variables.

    c. Figure 1 should be modified by using more distinct blocks, as the current shapes are not obviously different, and please mark the blocks that are not involved in this paper in the figure.

    d. In line 127, the paper states that 'the selected samples are physically meaningful', please further clarify this. Typically, wind drives the ocean waves, so does the sampling account for the wind-wave empirical correlation function? Or does it consider the wave-steepness characteristics?

    e. As mooring configuration is particularly site-specific, for instance, the water depth determines the total length, while the soil conditions decides the anchor selection. However, the sea state sampling is based on the non-site-specific training. In this sense, this paper applies a specific mooring design across 1000 sea states. How are these two principles validated simultaneously? Furthermore, how can it be ensured that the results are not mooring-specific?

    f. In case a specific environmental region is chosen, for instance North Altlantic (mentioned in line 133), what are the upper and lower bounds for all the five input environmental variables? Please provide more information on the input variable ranges.

    g. In table 3, please clarify whether the simulation time length corresponds to each test case with a single specific seed or represents the total simulation length for all six seeds.

    h. Line 210 states that the S-N curve are based on tests under mean loads remain 20% of the MBL. Does this mooring configuration meet this constraint? If not, since the mean loads influence fatigue, how can the application of the S-N curve parameters be validated? Does this paper consider the influence of the mean loads in the fatigue calculation in this study? Please provide further clarification on these.

    i. From line 220-235, the effect of corrosion is considered in the fatigue damage, by using an extended S-N curve as expressed in Eq2. How does the corrosion grade parameters used in the fatigue calculation for different phases, for instance, new, 10-year usage? Does this extended S-N curve consider the specific region or specific mooring design, since all coefficients are empirical estimated? Does this violate the non-site-specific sampling principle? Please further clarify these.

j. In line 216, the paper states that corrosion is simply based on a reduction of chain diameter. This is partially correct, since for life-time fatigue prediction, marine growth is critical, as it contributes to chain corrosion. The marine growth influences not only chain diameter, but also line mass, and drag coefficient of mooring lines, how does this paper consider these influences in fatigue prediction? If marine growth is ignored, what is the justification for considering the extended S-N curve? Furthermore, corrosion also reduces chain strength over time. How is this effect incorporated into the S-N curve, considering that the minimum breaking load (MBL) also decreases with time? Please provide further clarification on these aspects.

4. In the 'surrogate model' section:
   a. in line 250-260, since the computation time is compared between OpenFAST and surrogate models, please specify the version of OpenFAST.
   b. In line 258-265, for clarity, consider replacing 'the first subsection' with 'in Section 4.1' to provide a more precise reference.
   c. In table 4, consider restructuring the contents into the categories: 'Simplicity,' 'Handling Non-Linearity,' 'Accuracy,' 'Efficiency,' and 'Best Use Case' to provide a more distinct and structured comparison.
   d. In line 300, please clarify whether the random search method is used for all five surrogate models.
   e. In Figure 3, since the optimal hyperparameters are applied to the dataset again, should the workflow be structured accordingly, like this

5. In the 'environmental condition' section:
   a. In line 379, the paper states the water depth around 100m, how does this shallower water depth align with the FOWT model, which features the hydrodynamic properties and a mooring design for sites of 200 m? In Table 2, the anchor depth corresponds to the water depth of 200 m. The hydrodynamic properties as well as the mooring pretension significantly change with shallower water. Please clarify the modifications made for the Openfast simulation.
   b. In line 384, what is the wind direction?
   c. line 387 sees two dots at sentence end.
   d. In Figure 7, consider adding the peak values and the peak frequency for each environmental variable. It appears that the wave period is discrete rather continuous, please clarify this. Furthermore, specify the spectrum used for wave modeling and the turbulence model applied for wind modeling.

6. In 'result' section:
    a. In line 405, the paper states the results are only for line 1 with grade 3, please justify why this line at this corrosion grade is used to represent the long-term fatigue status of three mooring lines under wind-wave misalignments.
    b. In line 415, the paper states that 800 samples were used for training the model, while the remaining 200 samples were applied for comparison purposes. Please clarify how the selection process was performed. Consider provide a distribution of fatigue damage across all sea states, to ensure that no biased sea state was excluded from the training process.
    c. Please clarify how many iterations were perform to obtain these optimal hyperparameters and which method was used for each surrogate model, since c = 10 in Support Vector Regression appears a bit high.
    d. In line 426-430, the R2 result (Figure 8 & Table 9) indicates the first three surrogate models have limitations in handling non-linearity, while these limitations are known prior to the $R^2$ calculation, please justify the decision to use these models with their already-known limitations.

    e. Since the comparison between surrogate models and OpenFAST simulations depends on the selection of samples, please justify why overall fitness is considered more important than capturing high-damage cases, especially when the primary motivation is to monitor mooring failure due to fatigue damage. Additionally, please clarify the occurrence of these high-damage cases and justify why their significance is being overlooked. Furthermore, since none of the five surrogate models can capture high-damage cases, does this imply that the surrogate models are not suitable for predicting critical cases?

7. In the 'discussion' section:
    a. In lines 500–505, the paper highlights the novelty of applying wind-wave directional misalignment. However, no evidence is provided to demonstrate the significance of this variable, especially since only one line is considered in the results. Consider adding more data to demonstrate that this variable is indeed significant in mooring fatigue. In addition, please clarify the modification of hydro properties in the OpenFAST simulation to consider this directional misalignment, when reference FOWT only has one directional hydro input.

---

## Author Comment (AC2)

**Authors' response to comments on wes-2024-162**

We would like to sincerely thank the reviewers for their time and thoughtful feedback. Their constructive comments have helped us improve the quality and clarity of the manuscript and have also provided valuable directions for potential future work. Below, we provide detailed responses to each of the reviewers' comments. Comments are in *black italic* and responses in *blue*.

On behalf of all authors, Azélice Ludot

**Review #1**

Very interesting and promising work. I would only suggest a very minor editorial change. Specifically, instead of '...unrealistic failure likelihood predictions for mooring systems', I would characterised such likelihood as conservative.

We appreciate your suggestion regarding the phrasing of line 9 of the abstract. We agree that describing such likelihoods as *conservative* is a more accurate and constructive characterization, and we have updated the manuscript accordingly.

**Review #2**

**General comments**

The paper performs OpenFAST simulations of the 15-MW FOWT with a conventional mooring configuration under 1,000 sea states and applies five surrogate models to evaluate their predictions of hourly mooring fatigue damage. The best surrogate model, which has the lowest  $R^2$  values, is further used to estimate the uncertainty.

The paper is well-structured, and the topic of mooring fatigue monitoring is interesting, particularly with the use of surrogate models, which greatly improve efficiency. However, the novelty of the paper is not sufficiently demonstrated through the methods and results. The following comments are provided, with the hope of improving the quality of the paper.

**1. Introduction**

a. In line 39, the paper mentions a target of 60 GW by 2030. Please verify this with the latest literature, such as the Global Wind Report 2024 by the Global Wind Energy Council, which sets a target of 320 GW by 2030.

Thanks for flagging this! It has been updated using as recommended the figures from the Global Wind Report 2024.

b. The introduction does not clearly demonstrate the novelty of the proposed surrogate model in this study. What new functions or methods does the proposed surrogate model introduce? Or is it merely incorporating two more environmental variables—wave period and wind-wave misalignment—into tension prediction for digital-twin technology? How importance of these two factors in mooring fatigue?

Thank you for pointing out the lack of clarity in the introduction. As you correctly noted, the main novelty of this study lies in the inclusion of a broader set of environmental input variables, particularly wind and wave directions. The core idea behind this research is that considering only colinear wave and wind directions when computing failure rates may lead to conservative results. Therefore, we aimed to develop a surrogate model capable of predicting fatigue damage based on varying environmental conditions, including different wind and wave directions.

In addition, the fatigue damage computation incorporates other structural factors such as corrosion grade and mean load effects, to achieve more accurate estimates. We have revised the introduction to better emphasize these contributions. Furthermore, a new subsection has been added to the "Results" section, presenting a variance-based sensitivity analysis. This analysis quantifies the influence of each environmental input variable on the surrogate model's output, specifically assessing the importance of wind and wave directions on hourly fatigue damage. A potential extension of this work will be to investigate their influence on failure rates.

In lines 45–60, the paper discusses the high risk of mooring failures in the offshore oil and gas (O&G) sector, and the mitigation of these risks using tension sensors for real-time measurement. Furthermore, the literature cited in lines 60–65 mentions a platform motion-based method that addresses the issues associated with tension sensors.

However, the paper does not further elaborate on the novelty of the proposed approach. For instance, what is the specific importance of the surrogate model for condition monitoring? Why not use a GPS sensor directly instead of relying on a surrogate model?

The platform motion-based method mentioned by the reviewer is indeed effective in addressing the reliability issues associated with tension sensors. However, in the context of this work, the goal was to develop a versatile tool suitable for both real-time condition monitoring and longterm integration throughout the turbine's lifetime to assess failures and reliability. To support this objective, environmental conditions were selected as input variables. Regarding the use of GPS sensors, despite their reliability, post-processing is still required to extract damage-related information. Therefore, real-time monitoring and long-term integration still necessitate the use of surrogate models. These models, trained on databases of high-fidelity simulations, enable real-time or near-real-time fatigue damage estimation based on metocean measurements, without the need to run full OpenFAST simulations. This approach is particularly interesting for offshore wind farm monitoring, where computational resources are limited and rapid decision-making is essential for maintenance planning.

Since there is no interactive feedback for the operational or maintenance adjustments, but only post-processing of measurement data for fatigue prediction, how does the surrogate model or even digital twin technology offer a distinct advantage?

While this work does not yet implement interactive feedback for operational decision-making, it lays the foundation for such capabilities. Surrogate models integrated into a broader digital twin framework could, in future developments, support predictive maintenance, damage prognosis, and operational optimization in a computationally efficient manner.

The introduction lacks sufficient evidence to support the surrogate model's ability or digital twin technology to improve long-term mooring integrity in terms of fatigue. Fontaine et al. (2014), as the paper cited in line 55, found that 3 out of 29 mooring fatigue failures were caused by out-of-plane bending of chain links. In this case, in addition to tension values, the angles between two links are also crucial. However, the paper does not further discuss the capability of the platform motion-based surrogate model to predict the angles between two links. How did the surrogate models in literature (specially the platform motion-based method cited in line 60-65) or the proposed method for digital-twin technology, which in this paper incorporates two additional environmental variables, address this issue?

The effect of out-of-plane bending (OPB) on fatigue damage and failures has indeed been highlighted several times in the literature. However, this effect was deliberately excluded from the current study to simplify the problem. None of the studies cited in our manuscript considered OPB either. That said, we acknowledge the importance of this phenomenon and anticipate incorporating it in future work by post-processing the chain link angles and applying a hot-spot S–N curve for fatigue assessment. Traditional traction-based S–N curves, which do not account for stress concentrations, are insufficient to capture OPB effects accurately. Including OPB would increase the computational cost of building the training database, but as the reviewer rightly noted, it would lead to more accurate fatigue damage predictions. We appreciate your insightful question—this is certainly an area for future improvement. A sentence acknowledging this limitation has been added to the discussion section.

2. In the 'Reference system' section: please clarify the mooring pretension used in this study. Since varying pretensions influence the mooring stiffness and tension damage.

Since the mooring system used in this study is identical to the reference system described by Allen et al. (2020) in *Definition of the UMaine VolturnUS-S Reference Platform Developed for the IEA Wind 15-Megawatt Offshore Reference Wind Turbine* (National Renewable Energy Laboratory, https://www.nrel.gov/docs/fy20osti/76773.pdf), the pretension is assumed to be the same as reported in that reference. For clarity, a row indicating the computed pretension has been added to Table 2.

**3. In the 'Generation of the synthetic database' section:**

a. The paper describes OpenFAST as a 'high-fidelity' tool; however, its official webpage (https://openfast.readthedocs.io/en/main/) refers to it as a 'multi-fidelity' tool. Typically, CFD simulations are classified as 'high-fidelity'.

Thanks for the clarification. It has been modified in the text.

b. In line 116, the paper states that the sampling technique aims to avoid conditions that would never occur. Please clarify this sampling method further, especially in the context of using the 'non-site-specific training' approach for input variables.

The following three subsections under the main section titled "Generation of the Synthetic Database" describe the sampling procedures used for the non-site-specific training approach. The primary objectives are to avoid generating unrealistic environmental conditions and to preserve the correlations between variables. The procedure is summarized as follows:

i. Wind speed is assumed to be independent of all other variables. Its distribution is derived from 25 years of data collected across multiple sites within the study area.

- ii. All other environmental variables are assumed to follow uniform distributions within conditional bounds, which are informed by bivariate plots and standard industry practices.
- iii. The conditional bounds are defined to include 99.7%–99.9% of the observed data, ensuring that extreme events are not overrepresented in the synthetic samples.

We hope this clarification addresses the reviewer's comment. To further illustrate the methodology, we have added a reference to the "Environmental conditions" section and included a new example demonstrating how the sampling procedure is applied in practice.

c. Figure 1 should be modified by using more distinct blocks, as the current shapes are not obviously different, and please mark the blocks that are not involved in this paper in the figure.

These changes have been implemented.

d. In line 127, the paper states that 'the selected samples are physically meaningful', please further clarify this. Typically, wind drives the ocean waves, so does the sampling account for the wind-wave empirical correlation function? Or does it consider the wave-steepness characteristics?

The sampling procedure described in this section aims to preserve correlations between variables by defining conditional bounds for each variable based on a selected "driving" variable, thus avoiding unrealistic samples. The sentence line 127 has been extended. The study derives the conditional bounds not for empirical functions but from the data gathered across a variety of sites in the studied area.

e. As mooring configuration is particularly site-specific, for instance, the water depth determines the total length, while the soil conditions decides the anchor selection. However, the sea state sampling is based on the non-site-specific training. In this sense, this paper applies a specific mooring design across 1000 sea states. How are these two principles validated simultaneously? Furthermore, how can it be ensured that the results are not mooring-specific?

You raise an important limitation of the paper, which is addressed later in the discussion. While our aim was to develop a non–site-specific model, the current database is indeed mooring-system-specific. To overcome this limitation, design variables should be incorporated into the database inputs, enabling the surrogate model to account for both environmental and design parameters. This approach would allow the model to be adapted to different mooring systems across various sites and support reliability-based design optimization (RBDO). This is in the scope for future work.

f. In case a specific environmental region is chosen, for instance North Altlantic (mentioned in line 133), what are the upper and lower bounds for all the five input environmental variables? Please provide more information on the input variable ranges.

More information can be found in Table 6, which presents the case study of the North Atlantic.

g. In table 3, please clarify whether the simulation time length corresponds to each test case with a single specific seed or represents the total simulation length for all six seeds.

Thank you for highlighting this point, it is indeed not clearly stated. Each simulation, using a single specific seed, has a duration of 4200 seconds, including 600 seconds of transient time. Additional information has been added in Table 3.

h. Line 210 states that the S-N curve are based on tests under mean loads remain 20% of the MBL. Does this mooring configuration meet this constraint? If not, since the mean loads influence fatigue, how can the application of the S-N curve parameters be validated? Does this paper consider the influence of the mean loads in the fatigue calculation in this study? Please provide further clarification on these.

In this work, we do not use the conventional S–N curve formulation, which assumes a constant mean load. Instead, we adopt an extended formulation developed by Lone et al. (2021), which accounts for varying mean loads under different environmental conditions as well as the effects of corrosion. Therefore, the influence of both mean load and corrosion is explicitly considered in the fatigue calculations.

i. From line 220-235, the effect of corrosion is considered in the fatigue damage, by using an extended S-N curve as expressed in Eq2. How does the corrosion grade parameters used in the fatigue calculation for different phases, for instance, new, 10-year usage? Does this extended S-N curve

consider the specific region or specific mooring design, since all coefficients are empirical estimated? Does this violate the non-site-specific sampling principle? Please further clarify these.

The corrosion grades used in this study are defined in Lone et al. (2021) (see full reference in the article), specifically in Table 2. Each chain segment was assigned a corrosion grade to characterize its surface condition and the severity of corrosion observed. These grades were determined through visual inspection, using a scale from 1 (new or mildly corroded) to 7 (severely corroded). A detailed description of the corrosion categories is provided in Table 2 of Lone et al. (2021), with illustrative examples in Figure 1 of the same paper. We acknowledge the limitations of this grading system, particularly the subjectivity involved in visual assessments. However, it provides a practical means of accounting for key corrosion characteristics, such as the amount, depth, and location of corrosion pits, which are expected to impact fatigue life. Indeed, the corrosion grade is not uniformly applied to the entire mooring line; rather, it is specific to each segment. For example, chain segments near the fairlead typically experience more corrosion due to their position at the airwater interface. While this visual inspection method is not yet widely standardized in the industry, we anticipate that future developments, such as standardization efforts and increased field experience, will enable more objective corrosion grading, potentially based on pit depth, location, and chain age. We recognize this limitation and acknowledge that it introduces a degree of deviation from the principle of non-site-specific sampling.

j. In line 216, the paper states that corrosion is simply based on a reduction of chain diameter. This is partially correct, since for life-time fatigue prediction, marine growth is critical, as it contributes to chain corrosion. The marine growth influences not only chain diameter, but also line mass, and drag coefficient of mooring lines, how does this paper consider these influences in fatigue prediction? If marine growth is ignored, what is the justification for considering the extended S-N curve? Furthermore, corrosion also reduces chain strength over time. How is this effect incorporated into the S-N curve, considering that the minimum breaking load (MBL) also decreases with time? Please provide further clarification on these aspects.

Thank you for your comment on marine growth. In this work, we chose to disregard the influence of marine growth on the fatigue assessment; however, we acknowledge that it could have an impact on the results. The

extended S–N curve formulation was used to account for the influence of corrosion pitting. Your point regarding the variation of the minimum breaking load (MBL) is well taken. For simplification purposes, this aspect was not considered in the present study. Nonetheless, we are interested in exploring its potential impact in future work. A consideration on this point has been added into the discussion section.

**4. In the 'surrogate model' section:**

a. in line 250-260, since the computation time is compared between OpenFAST and surrogate models, please specify the version of OpenFAST.

A footnote with this information has been added.

b. In line 258-265, for clarity, consider replacing 'the first subsection' with 'in Section 4.1' to provide a more precise reference.

**Changed.**

c. In table 4, consider restructuring the contents into the categories: 'Simplicity,' 'Handling Non-Linearity,' 'Accuracy,' 'Efficiency,' and 'Best Use Case' to provide a more distinct and structured comparison.

Thank you for your suggestion to restructure Table 4. While we understand the potential benefits of such a structure, we found that implementing these categories would be difficult given the overlap of the factors being compared, and the time limitations to answer the comments. The current organization of the table was designed to clearly highlight the key distinctions and ensure the comparison remains straightforward. We appreciate your input, and while we decided not to make this change in the current version of the paper, we will certainly keep this in mind for future related work.

d. In line 300, please clarify whether the random search method is used for all five surrogate models.

The search method used for each model has been clarified for better understanding. Grid search is used in all models except with RF.

e. In Figure 3, since the optimal hyperparameters are applied to the dataset again, should the workflow be structured accordingly, like this [see figure].

Thank you for your observation. To clarify the workflow, we have revised the structure of Figure 3 to explicitly reflect this step.

**5. In the 'environmental condition' section:**

a. In line 379, the paper states the water depth around 100m, how does this shallower water depth align with the FOWT model, which features the hydrodynamic properties and a mooring design for sites of 200 m? In Table 2, the anchor depth corresponds to the water depth of 200 m. The hydrodynamic properties as well as the mooring pretension significantly change with shallower water. Please clarify the modifications made for the Openfast simulation.

In this work, the mooring system design has been maintained for a depth of 200 meters, even though the application sites are all 100 meters deep. We acknowledge that metocean conditions vary with depth, and this choice is an estimation based on the trade-off between selecting data from sites with existing or upcoming floating projects and maintaining a reasonable depth that aligns with the conditions for which the mooring system was originally designed. An acknowledgment of this has been added to the text.

b. In line 384, what is the wind direction?

The rose plot of the wind direction is the following (plotted with significant wave height).

c. line 387 sees two dots at sentence end.

Changed thanks!

d. In Figure 7, consider adding the peak values and the peak frequency for each environmental variable. It appears that the wave period is discrete rather continuous, please clarify this. Furthermore, specify the spectrum used for wave modeling and the turbulence model applied for wind modeling.

Thank you for the helpful suggestions. We have revised the manuscript to provide more detail on the environmental models used. Specifically, we now specify the wave spectrum model and the turbulence model used for wind modeling.

Regarding Figure 7:

- The figure presents bi-variate distributions of metocean data (e.g., wind speed, wave height, and wave peak period) based on 25 years of hourly data from eight selected sites obtained from ResourceCODE.
- ii. The wave peak period appears discrete because the source data itself is quantized at fixed interval bins, not because of a modeling assumption. We have clarified this in the caption and text.
- iii. The aim of the figure is to show variable interdependencies in real metocean conditions and to assist in deriving the wind speed distribution used in the analysis.

**6. In 'result' section:**

a. In line 405, the paper states the results are only for line 1 with grade 3, please justify why this line at this corrosion grade is used to represent the long-term fatigue status of three mooring lines under wind-wave misalignments.

Thank you for the comment. As stated, the surrogate model developed in this work is specific to a particular mooring line, corrosion grade, and design configuration. For this study, we selected Line 1 at corrosion grade 3 because it represents a **middle corrosion severity level** and corresponds to the **line we identified as the most critically loaded** under the studied conditions. This choice allows for a focused and meaningful analysis of fatigue behavior without the added complexity of end-of-life degradation. Due to time constraints, we limited the scope to a single surrogate model; however, future work could extend the methodology to other lines and corrosion grades to assess comparative fatigue performance.

b. In line 415, the paper states that 800 samples were used for training the model, while the remaining 200 samples were applied for comparison purposes. Please clarify how the selection process was performed.

Consider provide a distribution of fatigue damage across all sea states, to ensure that no biased sea state was excluded from the training process.

The split between training and testing was done randomly, and a clarification on this process has been added to the manuscript. We appreciate your feedback regarding the sea state distribution and will consider this approach in future work or papers. However, to limit the length of this paper, we have not included the verification in this study.

 Please clarify how many iterations were performed to obtain these optimal hyperparameters and which method was used for each surrogate model, since c = 10 in Support Vector Regression appears a bit high.

**100 combinations of hyperparameters have been tested.**

d. In line 426-430, the R2 result (Figure 8 & Table 9) indicates the first three surrogate models have limitations in handling non-linearity, while these limitations are known prior to the R2 calculation, please justify the decision to use these models with their already-known limitations.

We acknowledge that the first three surrogate models have inherent limitations in capturing non-linearity, as reflected in their lower  $R^2$  values (lines 426–430, Figure 8, Table 9). However, our decision to include and evaluate these models was intentional. Despite knowing their theoretical limitations, we believed it was important to quantify their actual performance on the dataset to:

- i. Establish a baseline for comparison against more advanced models;
- ii. Validate whether their simplicity could still offer acceptable accuracy in certain contexts;
- iii. Provide a complete and transparent assessment of all considered surrogate strategies.
- iv. This empirical evaluation allows us to objectively demonstrate the benefits and drawbacks of each model, rather than relying solely on theoretical assumptions.
- e. Since the comparison between surrogate models and OpenFAST simulations depends on the selection of samples, please justify why overall fitness is considered more important than capturing high-damage cases, especially when the primary motivation is to monitor mooring failure due to fatigue damage. Additionally, please clarify the occurrence of these

high-damage cases and justify why their significance is being overlooked. Furthermore, since none of the five surrogate models can capture highdamage cases, does this imply that the surrogate models are not suitable for predicting critical cases?

According to the literature, most fatigue damage over the turbine lifetime occurs during DLC 1.6, which corresponds to operational conditions. Therefore, storm conditions, when the turbine is parked, have less influence on overall fatigue. Based on this, we focused on accurately predicting the bulk of the data, leaving the extreme events (the tails) for future investigation. As a result, our surrogate models are not expected to perform well for extreme events, as these are underrepresented in the training database. However, we anticipate that this limitation can be addressed through future refinement of the dataset.

**7. In the 'discussion' section:**

a. In lines 500–505, the paper highlights the novelty of applying wind-wave directional misalignment. However, no evidence is provided to demonstrate the significance of this variable, especially since only one line is considered in the results. Consider adding more data to demonstrate that this variable is indeed significant in mooring fatigue. In addition, please clarify the modification of hydro properties in the OpenFAST simulation to consider this directional misalignment, when reference FOWT only has one directional hydro input

Thank you for highlighting the lack of evidence regarding the influence of wind and wave directions. To address this, we conducted a variance-based sensitivity analysis, which allowed us to quantitatively assess the impact of each environmental input variable on the predicted fatigue damage. This analysis provided clear insights into which parameters — including directional effects — contribute most significantly to the model outputs. Regarding the OpenFAST simulations, we would like to note that the latest hydrodynamic database released by NREL (accessible via the IEA-15-240-RWT repository on GitHub: IEAWindSystems/IEA-15-240-RWT) includes 36 hydrodynamic load cases that cover a wide range of wave directions, ensuring that directional dependencies are embedded within the simulation results used to train and evaluate the surrogate models.